# Beyond scores: A machine learning approach to comparing educational system effectiveness

Rogério Luiz Cardoso Silva Filho[1,2,3]*, Anvit Garg[1], Kellyton Brito[4], Paulo Jorge Leitão Adeodato[2], Martin Carnoy[1]

1 Graduate School of Education, Stanford University, Stanford, CA, United States of America, 2 Centro de Informática, Universidade Federal de Pernambuco, Recife, PE, Brazil, 3 Reitoria, Instituto Federal do Norte de Minas Gerais, Montes Claros, MG, Brazil, 4 Departamento de Computação, Universidade Federal Rural de Pernambuco, Recife, PE, Brazil

* rlcsf@stanford.edu

**Data Availability Statement:** Data relevant to this study are available at https://doi.org/10.7910/DVN/WEWDHL.

**Funding:** The author(s) received no specific funding for this work.

**Competing interests:** The authors have declared that no competing interests exist.

## Abstract

Studies comparing large-scale assessment data among educational systems have been an important tool for understanding the differences in how education is delivered worldwide. Many of these studies do not go beyond reporting average student scores in a particular educational system. A more unbiased analysis would avoid the simple use of gross performance and consider educational system contexts. A common approach is to estimate effectiveness by the residuals of parametric linear models. These models rely upon strong assumptions regarding the data-generating process, and are limited to handling extensive datasets. To address this issue, our paper provides a new approach based on machine learning models. The new approach is flexible, allows paired comparison, and is model-independent. An analysis conducted in Brazil verifies the suitability of the method to explore differences in effectiveness between Brazilian educational administrative units at the regional and state levels from 2009 to 2019. Our results are consistent with the existing literature, but the methodology produced a number of new findings that were not observed in studies using more traditional approaches.

## 1. Introduction

Educational improvement should be one of the top priorities for government development policies worldwide [1], since education is linked to social development, increases in labor productivity, improved general welfare, and economic growth. A primary aspect of this process is identifying more and less effective education systems and conducting in-depth investigations into the structures and processes that lead to better student performance. In this sense, Large-Scale Education Assessments (LSAs) tests have played an important role in promoting educational assessment from a national and international perspective [2]. LSAs provide a robust cycled process to analyze the ability of students to learn what educational systems intend them to learn or experts consider that they should be learning. Beyond data on student achievement,

LSAs collect contextual information about the educational system, including school and teacher data. This rich combination of data has produced considerable research that sheds light on how educational infrastructures, policies, and contexts operate and how related factors might contribute to educational outcomes [2, 3].

In this context, a useful analysis is to compare LSA achievement across different educational systems at some level of aggregation, such as schools, states, regions, or countries. However, this comparison is challenging. There is an implicit line of thought that education is shaped mainly by economic, political, and social forces defined by boundaries [4]. In other words, states, regions, or countries may differ in how they contribute to education due to differences in demographics, wealth, and beliefs. Therefore, an adequate comparison of these educational systems should consider their peculiarities [2]. Standard practices, which often depict simple raw rankings of mean achievement, could be biased and have been subject to significant criticism [2, 5, 6].

So far, efforts to make less biased comparisons have relied mainly on traditional statistics models based on a well-known effectiveness framework [7]. This framework compares observed student achievement with predicted student achievement given the contextual educational system particularities and estimating the so-called "effectiveness metric." Technically, this metric is often estimated by means of residuals of hierarchical linear models (HLMs) using the educational production function (EPF) [8], in which the LSA scores are the output and socioeconomic variables inputs. Recently, [9] used socioeconomic variables to build educational system clusters and, using network theory from the ecology field, assessed how the correlation between LSA score and socioeconomic background varies across countries to leverage insights into the success of local compensatory policies.

Nevertheless, these techniques are limited to handling extensive data available in LSAs and cannot thoroughly investigate hidden patterns present in these data. On the other hand, data mining (DM) and machine learning (ML) techniques are widespread and have enhanced data analysis tasks in different fields. Unlike traditional statistics, they are inductive and learn directly from the data without relying on strong assumptions about data distribution and a prior hypothesized relations between variables. Although DM and ML have already been used on LSA data, to our knowledge they are not yet used to compare educational system effectiveness.

To fill this gap, this paper provides a new benchmark approach to educational system effectiveness. It is also based on the traditional effectiveness framework. However, instead of computing within educational system variance by means of residuals of linear models, it focuses on the bias of ML classifiers. The main goal of this approach is to bring flexibility to the analysis, widening the range of potential comparisons of how different educational systems deliver education with few assumptions about the data. In other words, it takes advantage of the novelties of ML and DM to identify educational systems that exceed expectations to guide further investigations of practices and policies that can lead to educational improvement.

The study is general since it can be applied to most national and international LSAs at different levels of aggregation, such as states and countries. Also, it is flexible since it is model-independent and can be applied using any classifier. Lastly, it can derive insights into the estimated effectiveness from different perspectives. For instance, it is possible to compare whether an educational system contributes more than others in specific practices such as those linked to students or teachers. The approach was conducted to analyze Brazilian secondary education to identify differences in effectiveness across states in the national federation. Brazil is continent-sized, with a large economy, and has the second-largest secondary LSA worldwide [10]. Yet, it has received little attention in the international scientific community.

## 2. Related work

### 2.1 EDM in LSA

The application of DM and ML in education has been called educational data mining (EDM). Despite the evolution of frontiers in EDM, most research in this field has only concentrated on data from information systems or learning management systems from specific institutions [11]. See also recent review of EDM and its applications [12].

On the other hand, EDM is in its infancy when applied to LSAs. Existing works often advocate the novelty of EDM as an alternative framework to traditional statistical techniques due to their flexibility in handling multi-dimensional data and their strong data-driven approach [13–15]. This behavior relaxes the parametric assumptions to discover the data systematically without the need to impose a pre-defined hypothesis. Most studies use models of educational production functions (EPF). By modeling learning, they seek to predict student achievement based on variables collected in the LSA questionnaires [16]. Regression models are the most used approach, although classification has also been used, and statistical separatrices [15, 17], absolute thresholds [18] or unsupervised learning utilizing clusters has been used to perform the class labels [19]. As detected in previous studies, tree-based algorithms have been the most used technique for both regression and classification tasks [15].

The most common goal of EDM in LSA is identifying the factors related to educational performance by means of the relevance of features in achievement predictions instead of comparing performance. The relevance of features is often expressed by the inherent model explanations or post-hoc explanations techniques [20–23].

### 2.2. Comparing educational systems

Identifying effective education systems can be one important strategy for systematically improving education. Despite this, EDM has contributed in a limited way to evidence on the importance of certain inputs to increasing student outcomes. For example, in [24] the author analyzed differences in the school variables that predicted reading literacy in Chinese-speaking versus English-speaking countries using an international LSA. He built a specific model for each culture and compared the results. Other studies [18, 25, 26] used unique models to analyze differences in the predictive relevance of variables in student outcomes for each of the five Brazilian regions using a Brazilian LSA. Their results showed significant differences among regions, which suggested the need for further intranational analysis to understand why some variables were important for student outcomes in some regions and others were important in other regions.

Overall, although these studies identified potential differences among educational systems, most of these studies do not move forward on exploring these differences nor apply any effectiveness framework, which is commonly pointed to as the future direction of research and the one we will take in the paper.

### 2.3 Effectiveness

Most of the studies cited above use the measured performance of the student as a criterion variable for outcome without discussing effectiveness. There are many ways to express performance, and the "best" measure remains still subject to debate [20]. However, the effectiveness framework [7, 27], in which the relation between observed and expected performance given the peculiarities of the respective educational systems, has been widely accepted. Within this context, it is common to use two-steps [6, 22, 28, 29]. In this approach, school effectiveness is first estimated as the relative residuals of multilevel models when controlling only for socioeconomic features, thereby avoiding models skewing towards privileged schools. Second, the

computed measure in the prior step is modeled as a function of school characteristics to describe their relevance to school effectiveness. Also, using separate models, [20] compared the effectiveness of Hungry and Italy using a national LSA of each country. The authors used the economic data envelopment analysis (DEA) to compute the school's effectiveness. Finally, some studies use longitudinal data taking advantage of lagged scores. In this approach, the expected value is predicted by using early score (t-1) and school characteristics as predictors, next, the results are compared with the observed outcome (t) [29, 30].

In contrast with previous studies, the current approach does not assume any *a priori* distribution of the data and can be instanced by controlling for many variables, bringing more confidence to the results. Also, the current approach uses cross-sectional data, the most common form of data collected in LSA surveys.

## 3. Problem setup

Educational data where students are nested in classes, which are in turn nested in schools, cities, states, and so on, may not meet the independent and identical (i.i.d) property required for the most common statistical models [31]. In other words, the students may not follow the same data distribution at some level due to unobserved effects from this nested structure. These differences, if not controlled for, can bias aggregated analysis of LSA data. However, despite controlling for these biases, this paper seeks to measure and compare these bias differences. More specifically, the focus is on the differences in practices and policies of educational systems not observed on data within states, regions, and countries.

This scenario is illustrated in *Fig* 1. Following the EFP, $X = (x^1, \ldots, x^n)$ is a representative input vector of contextual features of the educational systems collected by LSA questionnaires

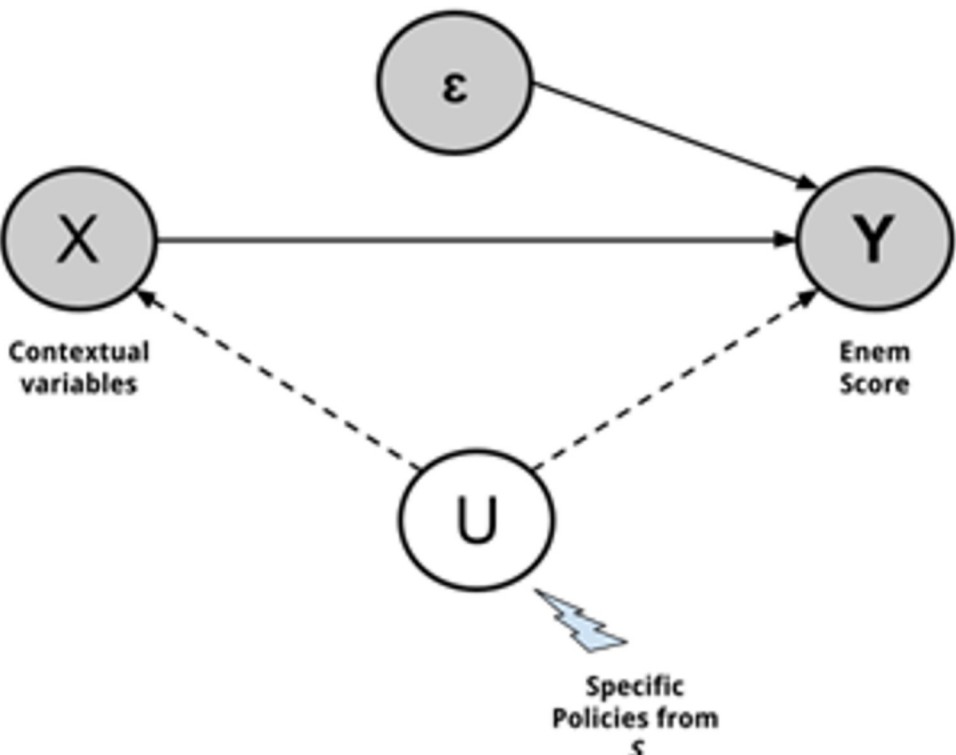

**Fig 1. Illustration of potential educational production function in which unobserved policies and practices from specific educational systems play a confounder role.**

and complementary data, which leads to the output $Y$, the LSA scores. $\varepsilon$ is a measure of unavoidable noise of the predictive model and is linear and equally distributed. Each educational system $s \in S$ can intervene differently through the effectiveness of policies and practice U $\notin X, Y$ to improve its educational system and is assumed in this paper to be the main source of bias in data. Therefore, $Y$ is a combination of the effects of $X$ and $U$, which can be described by the function $\hat{f}(X, \mathrm{U} + \varepsilon) = \hat{Y}$. Hence, considering that the unobserved confounder effects of $U$ may vary across $S$ ($U^S$), as an attempt to shed light on the differences in effectiveness across educational systems, the current work aims to leverage insights to find out to what extent this happens and where it remains the same.

## 4. Method

To compare the effectiveness of educational systems in various Brazilian regions, we use an educational effectiveness framework in which observed student performance averaged at the school level is contrasted with expected performance based on student, family, and school input characteristics [7, 27]. More specifically, we are interested in the differences in performance across subpopulations (administrative units) $S$ of an efficient function $\hat{f} = f(X, U)$ that predicts school student achievement outcomes when they are trained viewing all train data and tested across subpopulations defined by $S$.

It would be expected that performance would not change significantly if $S$ were drawn for the same distribution and $X$ a sufficient representative feature vector of educational systems. In other words, considering $X$ representative enough, significant differences in performance could be a proxy to shed light on the role of $U^S$; that is, in the effectiveness of the educational system in delivering education above or below that predicted by observable variables $X$. Furthermore, considering $X \cup U$, the role of $U$ may have different interpretations while $X$ varies its composition since all variables that are not present in $X$ will be present in $U$. For instance, if $X$ accounts only for a representative set of socioeconomics variables, the comparative analysis of $U$ will be related to differences in practices regarding all other variables other than student socio-economic characteristics.

It should be noted that the unobserved efforts compared in this approach are fundamentally different from the residual terms in linear regression equations. The residuals, or *random effects*, by definition, are uncorrelated with the other independent variables and should not be decomposed at the cost of misleading the physical reality [32, 33]. Otherwise, as defined in the problem setup, effectiveness is a part of the phenomenon, which is omitted but responsible for variations observed in the data. And although they cannot be measured accurately, it is important to assess them and their relationship with other variables quantitively and qualitatively.

### 4.1 Slicing analysis

To understand whether a model performs poorly on certain parts of the data, slicing analysis (SA) [33] has been used in research questions such as fairness [34–36] and business analytics [37]. In this paper, we bring in SA to test educational effectiveness and establish the comparison of performance as a paired hypothesis test with the following null ($H_0$) and alternative ($H_a$) hypotheses:

$$H_0 : \omega(\hat{f}, s) \leq \omega(\hat{f}, s')$$

$$H_a : \omega(\hat{f}, s) > \omega(\hat{f}, s') \tag{1}$$

Where $\omega$ is the performance of, $\hat{f}$ when tested in a specific educational system $s$, and $s'$, another educational system that is not present in $s$. To address the issue of evaluating the performance of a model $\hat{f}$ in different educational systems, a straightforward alternative is to use binary classifiers. This allows for easy systematic testing of paired comparisons, as outlined in (1). Additionally, this approach makes the analysis simpler and more accessible to most machine learning algorithms. Furthermore, by using a data-dependent strategy based on quantiles to set thresholds, it becomes easier to compare results across different datasets over time, as the classification is based on the relative position of data points within the dataset, rather than absolute values. To account for $\omega$ we use the False Negative Rate (FNR) and True Positive Rate (TPR). Therefore, considering the following:

- $X$ a representative vector of contextual features of the educational systems

- $\hat{f}$ a well-performed model across all educational systems

- $s$ and $s'$ large enough with balanced classes regarding $\hat{y}$.

$H_0$ is rejected when $\hat{f}$ tends to classify observations as systematically different between $s$ and $s'$. Let FNR be the measure $\omega$. A higher $\omega$ for $s$ reveals a tendency of $\hat{f}$ in misclassify high achievement (i. e., $\hat{y} = 1$) as low achievement (i. e., $\hat{y} = 0$) more in $s$ than in $s'$. In other words, high-achieved observations in $s$ generally have characteristics closer to the low-achievement observations present in the whole train data when compared to $s'$. Hence, the unobserved effects $U^s$ seems to be higher than $U^{s'}$, indicating that $S$ is relatively more effective in alleviating their unprivileged contextual schools' conditions. On the other hand, $\omega$ referring to FPR would indicate that observations of $s$ tend to be "overestimated", so $s$ do relatively less than expected regarding $s'$, and are less effective. However, if $U$ is uniform across $S$, $\omega$ should not be significant, then $H_1$ is accepted; thereby $s$ and $s'$ are somewhat equally effective. To test the hypothesis, a two-proportion Z test was used.

$$Z = \frac{\delta^s - \delta^{s'}}{\sqrt{\delta_s(1 - \delta_s)\left(\frac{1}{n_s} + \frac{1}{n_{s'}}\right)}}$$

Where $\delta$ is the proportion of misclassified observations (false negative to FNR and false positive to FPR) and $n$ the total of negative observations to FNR and positive to FPR.

Characterizing effectiveness: to understand the differences in effectiveness regarding a specific set of variables, an interesting exercise is to compare the same educational systems $s \in S$ when the model is trained with the *full* set of variables $X$ and when a specific set $x^n \in X$ is taken out. In this exercise, a uniform performance degradation across $S$ is expected. Otherwise, $x^n$ plays different roles across S. Therefore, considering $x^n$ as having a low correlation with the remaining variables $(X - x^n)$, it is possible to analyze the bias introduced or alleviated by $x^n$ in $S$. This performance difference can be tested by each $s \in S$ using a chi-squared McNemar's statistical test [38] and further compared across $S$ using the $\Delta FNR$ and $\Delta FPR$ by using (1).

$$Mc_s = \frac{(|n_{01} - n_{10}| - 1)^2}{n_{01} + n_{10}},$$

where, $n_{01}$ is the number of misclassified observations by the first model but not by the second and $n_{10}$ the number of misclassified observations by the second model but not by the first. The value, -1, in the numerator is a correction term.

## 4.2 Empirical data

The data utilized in this paper is from the Brazilian National Secondary School Exam (Exame Nacional do Ensino Médio–ENEM) [39], the second-largest secondary LSA globally [10]. ENEM released data containing student scores in the five ENEM tests (math, languages, natural sciences, human science, and an essay) and socioeconomic information on students, which were collected through extensive student questionnaires. Together with the national *school census*, which details the conditions of Brazilian schools—from physical infrastructure to teacher characteristics—the data are a valuable source of information about Brazilian secondary education. Both databases are publicly available and are released annually. The period covered is from 2009, when ENEM was reframed to make the scores comparable over time, to 2019, the last year currently available for both databases.

## 4.3 Scope definition

This study focuses only on public secondary schools, which enroll the majority of Brazilian secondary school students. The dataset refers to a total in all years of more than 40 million students in tens of thousands of schools across the country. However, only a fraction of these students is in the last year of secondary education, when students take the ENEM test. In addition, we used a set of filter criteria to define the scope of the study and these further reduced the number of students composing the sample in our analysis:

- As the school "id" will be the primary key for ENEM and *school census* dataset fusion, all students who do not attend a school that has been identified were removed.

- Students were not included if they were not in the last year of municipal or state public secondary schools.

- Students were not included if they did not follow a regular curriculum.

- As a double-check, students not in the most probable age range meeting criteria 1 and 2 (17–19 years old) were also eliminated.

- In order to obtain a critical mass, only schools with ten or more students were selected.

- To ensure that all schools had at least a minimum infrastructure to function, schools with no electric energy, sanitation, or piped water were not included.

The same criteria were applied for each year of the dataset. *Table* 1 presents the number of students and schools in public education before and after limiting the dataset. As the teacher table is present only in the *school census*, which considers every teacher from every school in Brazil (including non-secondary), only the final number of teachers related to schools at the ENEM level is presented. All variables were transformed to the school level, which is the level of the analysis (decision grain).

## 4.4 Data preprocessing

The data have gone through several changes over time. As an illustration, there were 293 variables in the ENEM questionnaire in 2009, while in the following year, 2010, only 57. Added to this difference in the number of variables collected, there were also changes related to the variables representation, such as 1) features were binary for some years and quantity for others; 2) categories were represented by numbers in some years and by strings in others; and 3) categorical features with $i$ options were transformed into $i$ binary features for different years.

**Table 1. Number of observations regarding public schools for each year.**

| Year | ENEM—before | | Final dataset–after | | |
|---|---|---|---|---|---|
| | Schools | Students | Schools* | Students | Teachers |
| 2009 | 22,696 | 1,173,419 | 15,413 | 631,604 | 492,584 |
| 2010 | 22,496 | 1,096,483 | 17,957 | 774,937 | 553,909 |
| 2011 | 22,274 | 1,200,923 | 19,584 | 911,309 | 602,292 |
| 2012 | 22,240 | 1,201,036 | 20,120 | 954,009 | 614,749 |
| 2013 | 21,085 | 1,293,786 | 21,085 | 1,049,134 | 649,652 |
| 2014 | 22,846 | 1,344,736 | 21,383 | 1,070,778 | 671,383 |
| 2015 | 21,843 | 1,310,702 | 21,843 | 1,128,398 | 701,424 |
| 2016 | 24,217 | 1,523,161 | 22,476 | 1,192,812 | 709,619 |
| 2017 | 24,102 | 1,433,841 | 21,692 | 1,078,598 | 676,138 |
| 2018 | 22,729 | 1,151,207 | 20,617 | 920,310 | 615,011 |
| 2019 | 21,069 | 933,988 | 18,149 | 777,589 | 586,718 |

* Decision grain

It is important to standardize the data to overcome these issues and to allow us to compare findings over the years. First, we select only variables present in all waves. Next, the data were standardized in regard to content and meaning. A variable with less information was used as a reference for mapping the others. For instance, if a variable was binary in one year and multiple categorical in the others, the binary version was adopted for all years. The income features were normalized using a contemporary minimum wage. The variable related to the use of technological devices were individually treated. For example, before 2019, the available information on technology devices at school were measured by just one variable (student's computer), in 2019, the questions also asked about notebooks and tablets, which were unified. The missing values were analyzed separately, since there were not many of them, and were given the most probable value. Alternatively, the mean of the non-missing values was used for those that did not have a clear explanation. To reduce the influence of outliers, all numerical features were normalized for each year separately, using the $\alpha$-winsorized values of the distribution ($\alpha/2 = 0.025$ at each tail) as their minimum and maximum.

As a strategy to enhance the discriminant power of data, some variables frequently brought to the fore in discussions on the quality of secondary education [40], which were not initially present in the databases, were created. These were: a) appropriate training of faculty members (measure by the ratio of teachers with the right background for the subject they teach;) b) the number of jobs held by the teacher (the average number of schools in which teachers work); c) faculty pedagogical training (the proportion of faculty with pedagogical training); d) faculty *DOMAIN* (the proportion of teachers in the school teaching in each *DOMAIN* covered on the ENEM); and e) faculty workload (the ratio of teachers to the number of subjects covered in the school). In the end, as mentioned above, 41 input variables compose the final variable set.

All data were averaged to the school level, and variables from students and teachers were aggregated. Overall, the central tendency for each school, such as median and mode, was adopted. For categorical variables with an ordinal relation, such as faculty and parent's education, domain knowledge was incorporated by using the number of years of education as a weight in the average calculation. Higher weight was given to advanced degrees such as Ph.D. and lower weight to lower degrees such as B.Sc. The variables were normalized to obtain a normal distribution and fall within a range of 0 and 1.

The chosen variable to indicate the outcomes was the arithmetic mean of the students within schools in all areas of knowledge covered in the tests. The median was adopted as a

threshold for labeling schools as high and low achievement, balancing the distribution of classes across the administrative units and increasing the range of potential analysis. The final dataset, along with the descriptive statistics, can be found in S1 Appendix.

## 4.5 Modeling and evaluation

This paper considers three classifiers widely used in ML literature: logistic *regression*, *random forest*, and *adaboosting*. The estimates were performed taking into account state and region identifiers. State is the straightforward unit of comparison regarding the Brazilian educational system [4, 41]. Brazilian regions are commonly compared in educational literature and may also provide insights into variation in effectiveness at a higher level of aggregation than the state with some historical justification for doing so. Regions have historical meaning and states within regions have many economic and social similarities. For instance, educational policy borrowing is more likely to find smoother implementation across neighboring states with similar social conditions.

To carry out the classifiers, we considered a leave-one-group-out cross-validation (LOOCV) setting. Specifically, the whole dataset sample was used and schools were classified above and below the sample median based on the average school scores for each state (and region). In other words, the algorithms iterated through the entire dataset, and in each iteration, one state (and region) is used as the baseline test for the model trained in those that remain. This setting supports the evaluation of the variance of the model, raising confidence that the model can generalize across the whole data. Therefore, the learned parameters capture something about the whole country and not just a portion of it. Predicted values for each state (and region) were then put together from the iterations to evaluate the overall prediction performance. For more reliable estimates aiming for less noise in the metrics of interest (FNR and FPR), each classifier was run five times, and the results were averaged. In the end, as the focus is to analyze the parity error performance instead of models, the classifier (logistic *regression*, *random forest*, and *adaboosting*) that produced the highest F-measure (F1) for each combination of features was chosen.

The models are evaluated for different sets of predictors to understand where effects are concentrated. In specific terms, the predictor variables were grounded in four categories: *non-actionable* (*race* and *gender*), *student socioeconomics*, *school infrastructure*, and *teacher information*. They are also assembled in all possible combinations. Therefore, 45 models were carried out for each year at the regional and state levels. The models were carried out with the default parametrization from the Scikit-learn Python library. [42] as the primary focus was on evaluating the proposed method rather than obtaining the optimal model fit. The code of all experiments is available at https://github.com/rogerioluizsi/EduEffectiveness.

## 4.6 Traditional models comparison

The current estimates are compared with traditional HLM at the state level. Eq 2 is used to extract the matrix of state's random effects as a measure of effectiveness.

$$Y_{js} = X\beta + Zs + \varepsilon \tag{2}$$

Where $Y_{js}$ is the ENEM score, $X$ is the *full* set of variables, Z is the set of *random effects* for each state *s* and is the parameter of interest, and $\varepsilon$ is the random error. The mean of Z is included in the estimates of $\beta$, the fixed effects. The matrix of *random effects Z* is the variation from its mean $\beta$ and represents the unobserved part of the model.

It is worth noting that there are great differences in the definitions of the methods as well as in how they derive results. The HLM computes a measure of effectiveness (*Z*) simultaneously

for all states regardless of whether some are not comparable due to the large contextual differences. In our study using ML methods the estimates are more flexible and paired established. Also, HLM assumes $Z$ as normally distributed and uncorrelated with $X$, while this study makes no assumptions about the data distribution. Nevertheless, similarities between the method's results in the real-world data are great expected, especially regarding the administrative units assigned as higher and lower effective. This similar behavior would support even more the suitability of the proposed ML approach to measure and compare educational effectiveness. In the same way, differences can shed light on the limitations and advantages of both methods.

## 5. Results

This section presents the results of applying the proposed approach in the context of Brazilian public secondary education. To make more accessible the interpretation of graphs and the results, *Table 2* depicts the Brazilian states, including Federal District, with their abbreviations and how they are organized in regions.

### 5.1 Model evaluation

A model with consistent assumptions and a good predictive quality closes its estimated function to the law of nature [43]. In our case, to increase confidence in our estimates, it is crucial

**Table 2. Abbreviation of Brazilian regions and states.**

| REGION | STATE | ABBREVIATION |
|---|---|---|
| MIDWEST (MW) | Distrito Federal | DF |
| | Goiás | GO |
| | Mato Grosso | MT |
| | Mato Grosso do Sul | MS |
| NORTH (N) | Acre | AC |
| | Amapá | AP |
| | Amazonas | AM |
| | Pará | PA |
| | Rondônia | RO |
| | Roraima | RR |
| | Tocantins | TO |
| NORTHEAST (NE) | Alagoas | AL |
| | Bahia | BA |
| | Ceará | CE |
| | Maranhão | MA |
| | Paraíba | PB |
| | Pernambuco | PE |
| | Piauí | PI |
| | Rio Grande do Norte | RN |
| | Sergipe | SE |
| SOUTH (S) | Paraná | PA |
| | Santa Catarina | SC |
| | Rio Grande do Sul | RS |
| SOUTHEAST (SE) | Espírito Santo | ES |
| | Minas Gerais | MG |
| | Rio de Janeiro | RJ |
| | São Paulo | SP |

**Table 3. Average and standard deviation of AUC for each subpopulation (states and regions) over the years.**

|  | 2009 | 2010 | 2011 | 2012 | 2013 | 2014 | 2015 | 2016 | 2017 | 2018 | 2019 |
|---|---|---|---|---|---|---|---|---|---|---|---|
| **BRAZILIAN STATES** | | | | | | | | | | | |
| **MEAN** | 0.66 | 0.78 | 0.78 | 0.76 | 0.78 | 0.78 | 0.76 | 0.77 | 0.76 | 0.75 | 0.76 |
| **SD** | 0.02 | 0.01 | 0.04 | 0.05 | 0.04 | 0.04 | 0.02 | 0.03 | 0.01 | 0.02 | 0.03 |
| **BRAZILIAN REGIONS** | | | | | | | | | | | |
| **MEAN** | 0.67 | 0.74 | 0.73 | 0.76 | 0.77 | 0.78 | 0.74 | 0.77 | 0.76 | 0.77 | 0.74 |
| **SD** | 0.04 | 0.02 | 0.04 | 0.03 | 0.02 | 0.02 | 0.03 | 0.02 | 0.02 | 0.02 | 0.04 |

that the predictive ability of the models is consistent across the dataset. In other words, a low variance of model performance will ensure that the model error is related mainly to the biases, which is our parameter of interest. Thus, evaluating the models' performance across the country, whether at the state or regional level, is important. *Table 3* shows the Area under ROC curve (AUC) average for each year and its standard deviation for the cross-validation process. Also, *Fig 2* presents the probability scores from the cross-validation procedure to the year 2019, showing that schools across the country have a similar likelihood of being classified as high achievers (y = 1) irrespective of the region. An exception is the South which does not have schools in the first decile. Nevertheless, the similar estimates for all other deciles across all regions, suggests the model is well calibrated throughout the country. These results are consistent with the state-level results in 2019 and the previous years.

## 5.2 Comparison

Despite the good performance of models, there is a very notable difference in LSA scores across the country. *Fig 3* illustrates the fraction of positive class (y = 1) at the regional level

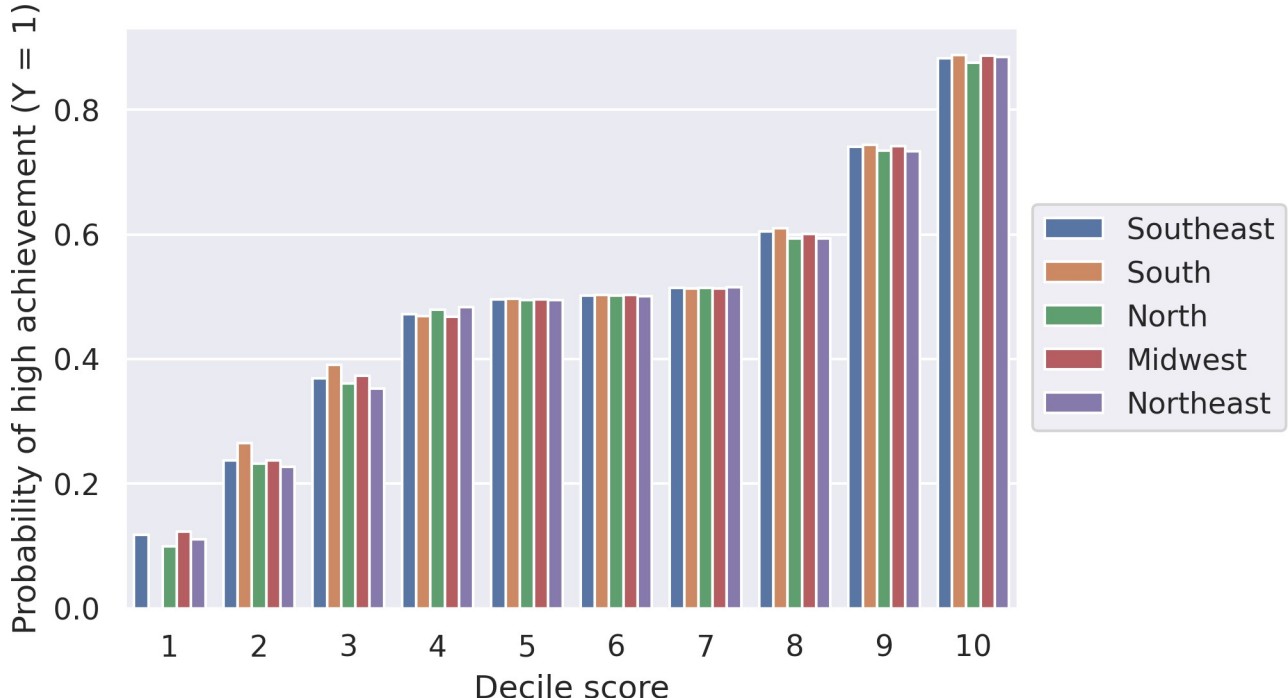

**Fig 2. The probability of a school being classified as high-achieving for the full set of variables model from 2009–2019, separated by Brazilian region and deciles, as determined by the best model evaluated using the F-1 measure.**

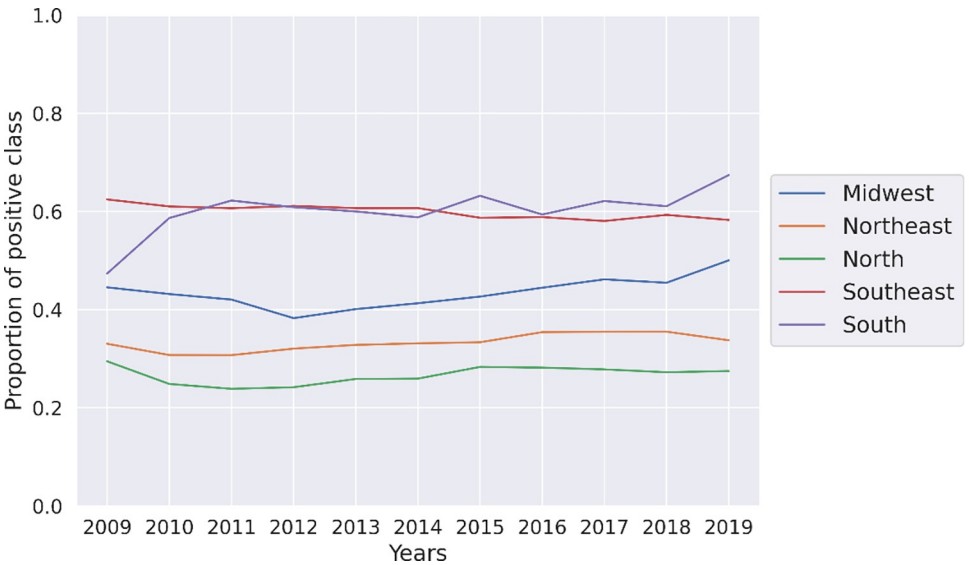

**Fig 3. Proportion of schools which scored above the median (positive class) by Brazilian regions from 2009 to 2019.**

from 2009 to 2019, while *Fig 4* shows the evolution from 2009 to 2019 at the state level. While the South and Southeast regions present a higher number of schools above the median, the North and Northeast present the worst results during the time studied.

This class and size unbalancing challenge our comparison since subpopulations with a lower concentration of positive class tend to achieve higher FNR and lower FPR, while those with the higher positive class do the opposite. However, the proposed approach was paired established, and since it does not have any identification of subpopulations in the train data, which could lead the algorithm to classify schools into the majority class, comparisons among

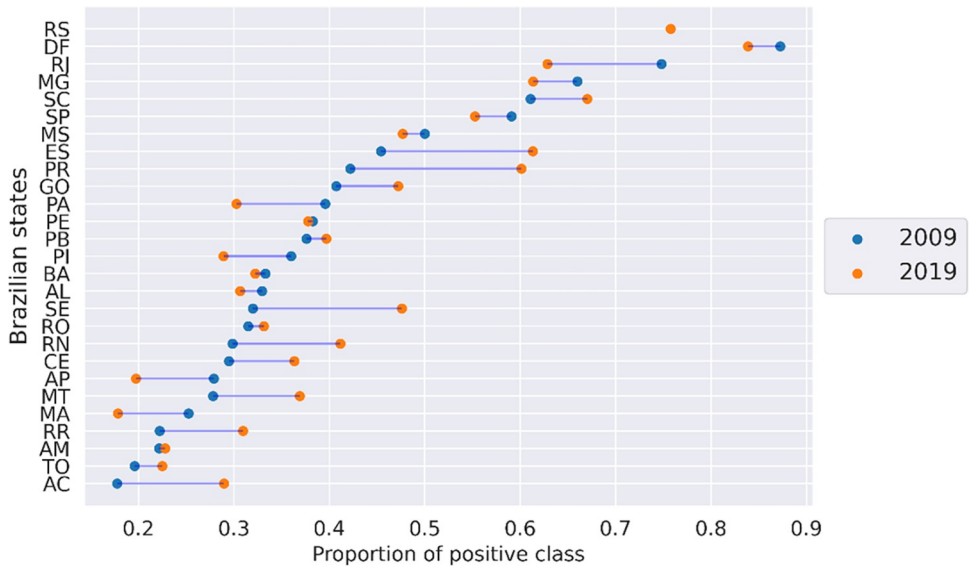

**Fig 4. Proportion of schools which scored above the median (positive class) in 2009 and 2019 by states.**

**Table 4. FNR and FPR by regions for each combination of features.** Also, the overall F1 -measure and AUC to the well-performed model M.

| Features | FPR | | | | | FNR | | | | | F1 | AUC | |
|---|---|---|---|---|---|---|---|---|---|---|---|---|---|
| | S | SE | ME | NE | N | S | SE | ME | NE | N | BR | BR | M |
| Full | **0.83** | 0.42 | 0.49 | 0.01 | 0.14 | 0.07 | 0.26 | 0.21 | **0.85** | 0.51 | 0.71 | **0.80** | RF |
| School | 0.31 | 0.17 | 0.36 | **0.63** | 0.50 | **0.64** | 0.73 | 0.51 | 0.24 | 0.30 | 0.43 | 0.48 | AB |
| Student | **0.72** | 0.54 | 0.52 | 0.00 | 0.16 | 0.07 | 0.21 | 0.20 | **0.88** | 0.48 | 0.71 | 0.79 | LR |
| Teacher | 0.52 | 0.17 | 0.30 | 0.71 | 0.37 | 0.5 | 0.76 | 0.49 | 0.22 | 0.45 | 0.44 | 0.48 | AB |
| Non-actionable | 0.85 | 0.34 | 0.30 | 0.21 | 0.11 | 0.07 | 0.39 | 0.41 | 0.53 | 0.68 | 0.67 | 0.74 | AB |
| School + Student | **0.77** | 0.44 | 0.58 | 0.02 | 0.25 | 0.08 | 0.25 | 0.17 | **0.79** | 0.40 | 0.71 | 0.79 | AB |
| School + Teacher | 0.5 | 0.16 | 0.32 | **0.62** | 0.38 | 0.49 | **0.72** | 0.55 | 0.25 | 0.41 | 0.46 | 0.51 | AB |
| School + Non-actionable | **0.89** | 0.29 | 0.30 | 0.20 | 0.13 | 0.04 | 0.41 | 0.40 | 0.54 | **0.66** | 0.67 | 0.74 | AB |
| Student + Teacher | **0.76** | 0.52 | 0.55 | 0.01 | 0.17 | 0.07 | 0.21 | 0.18 | **0.84** | 0.52 | 0.71 | 0.79 | LR |
| Student + Non-actionable | **0.77** | 0.46 | 0.54 | 0.03 | 0.19 | 0.07 | 0.23 | 0.17 | **0.76** | 0.41 | **0.72** | **0.80** | AB |
| Teacher + Non-actionable | **0.91** | 0.24 | 0.25 | 0.25 | 0.07 | 0.05 | 0.48 | 0.41 | 0.47 | **0.73** | 0.65 | 0.73 | AB |
| School + Student + Teacher | **0.80** | 0.48 | 0.52 | 0.02 | 0.13 | 0.06 | 0.22 | 0.20 | **0.78** | 0.55 | **0.72** | **0.80** | LR |
| School + Student + Non-actionable | **0.84** | 0.45 | 0.52 | 0.04 | 0.21 | 0.04 | 0.23 | 0.20 | **0.77** | 0.43 | **0.72** | **0.80** | AB |
| School + Teacher + Non-actionable | **0.81** | 0.32 | 0.23 | 0.13 | 0.09 | 0.13 | 0.41 | 0.00 | 0.64 | **0.74** | 0.65 | 0.74 | RF |
| Student + Teacher + Non-actionable | **0.91** | 0.51 | 0.50 | 0.00 | 0.12 | 0.01 | 0.22 | 0.22 | **0.88** | 0.57 | **0.72** | 0.79 | LR |

similar subpopulations can bring high confidence regarding the unobserved differences between them. This comparison assumes a significance level of 0.05.

**5.2.1 Brazilian regions.**   *Table 4* describes the False Negative Rate (FNR) and False Positive Rate (FPR) slicing by all regions for all combinations of variables to the year 2019. Also, the AUC to the model (M) that achieves the best F1. Observing the *full* model, the FNR metric highlights lower model confidence in Northeastern and Northern schools than in the others. The above-median Northeastern schools are overly classified as below the median, with a rate of 85%. This difference is statistically significant when compared with all other regions, including the North, which has a similar fraction of positive class (*Fig* 3). It suggests that Northeastern schools have a higher relative number of schools that score above the median with similar patterns as those that score below nationally. In other words, the Northeastern regions seem to have more effective policies when controlling for all possible contextual variables.

In attempting to explain the relationship of educational politics and policies to school achievement, a possible analysis is to consider that all variables beyond demographics are endogenous to government efforts. Hence, *Fig* 5 illustrates the comparison of a model that accounts only for demographic information (*student + non-actionable*) to the regional level for FPR (a) and FNR (b) in the year 2019. Each cell color represents the ratio between the states from the axis *y* by its respective axis *x*. The regions is in descending order, and clearer cells suggest larger differences between regions, while crossed-out cells represent those statistically significant in a two-sided Z-paired test.

As in the case where we consider the *full* model, there are significant differences across the country for both metrics in all paired comparisons. The Northeast is the most effective region, and the South is relatively the least effective. This pattern has remained consistent over the years. *Table* 5 shows the ratio of the FNR of all regions over time using the South as a reference. The Northeast has the higher value for the whole period, with 2014 as the year of highest differences in the effectiveness of governments when compared with the South region.

The consistency of the *full* models results to the demographic model (*student + non-actionable*) reinforces the lack of discrimination power of *teachers* and *school* features across the country. Observing Table 4 logistic *regression*, *random forest*, and *adaboosting*, both *teacher*

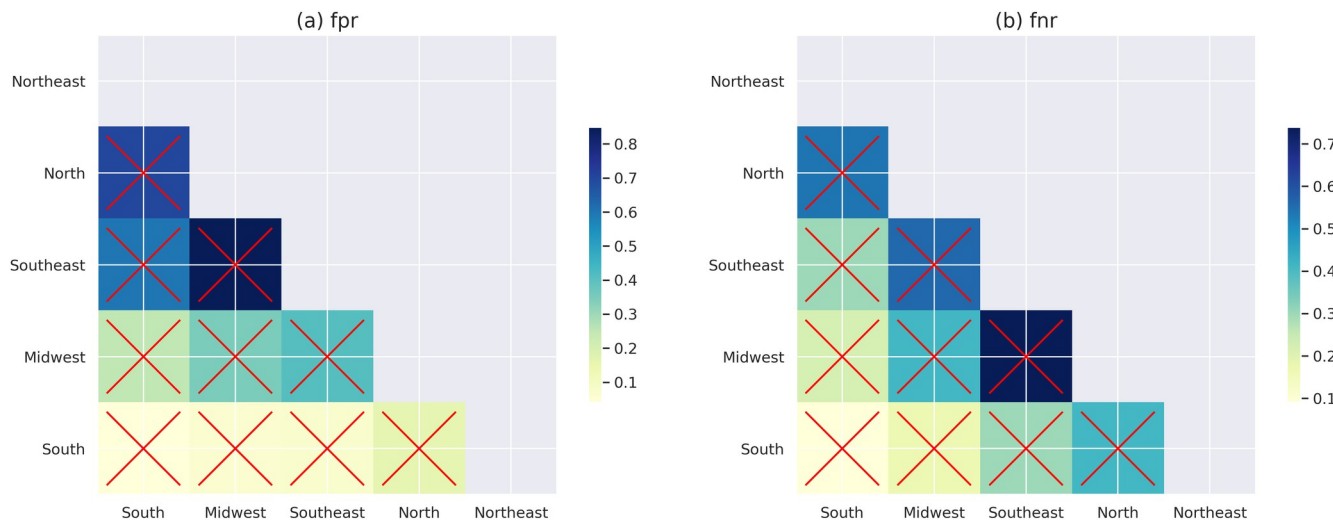

**Fig 5. Illustration of differences in error predictions for the demographic features set (*student + non-actionable*) in the year 2019 for the Brazilian regions.** Each cell represents the ratio of regions placed on axis x by those on axis y. clearer colours represent higher absolute differences while the crossed line represents whether they are statistically significant. (a) fpr and (b) fnr.

and *school* features models do not reach enough quality (AUC and F1 below 0.5), and their FNR and FPR cannot derive useful information. Also, there are no statistical differences in the changes (Δ) from the *full* models to the models without *teachers* (*students + teacher + non-actionable*) and without *schools* features for all regions. On the other hand, Table 4 also indicates the higher predictive utility of the demographic features set (*student + non-actionable*). Specifically, those included in the *student* set, *family income* and *parent's education*. Indeed, this set of features almost reached the same quality (F1 and AUC) and the same biases (FNR and FPR) as the *full* model, highlighting the usefulness of the other set of features when comparing Brazilian regions using cross-sectional data. Also, a strong correlation exists between the *non-actionable* and *student* variables. This was because *non-actionable* has a considerable discriminant power when used alone (AUC of 0.74 and F1 of 0.67) but increases the model performance marginally when combined with student features.

**5.2.2 Brazilian states.** This section describes the effectiveness benchmark at the state level. *Fig 6* illustrates the FNR for all states within regions with similar overall scores for 2019. The states are placed in descending order regarding effectiveness in the axes. The cell color represents the ratio of axis x by axis y. The clearer the cell (close to 0), the bigger the differences between states which are crossed when the difference is statically significant. *Fig 6(A)* confirms previous regional analysis results, and although they perform similarly, almost all states from the Northeast are more effective when compared with the North. The exception is PA which is

**Table 5. Ratio of FNR for each region compared to the FNR of the South region over time.**

|  | 2009 | 2010 | 2011 | 2012 | 2013 | 2014 | 2015 | 2016 | 2017 | 2018 | 2019 |
|---|---|---|---|---|---|---|---|---|---|---|---|
| **Midwest** | 5.0 | 5.2 | 6.75 | 8.00 | 6.00 | 9.00 | 6.637 | 5.00 | 8.00 | 6.00 | 2.43 |
| **North** | 12.8 | 11.00 | 12.75 | 16.67 | 18.33 | 26.00 | 19.67 | 12.5 | 17.33 | 18.00 | 5.86 |
| **Southeast** | 7.6 | 7.20. | 8.5 | 10.67 | 10.67 | 16.5 | 9.33 | 6.25 | 8.67 | 8.33 | 3.29 |
| **Northeast** | 14.8 | 15.2 | 21.00 | 27.33 | 27.33 | 43.00 | 29.33 | 22.25 | 29.33 | 29.33 | 10.86 |

Note: All differences are statistically significant at the p<0.05 level.

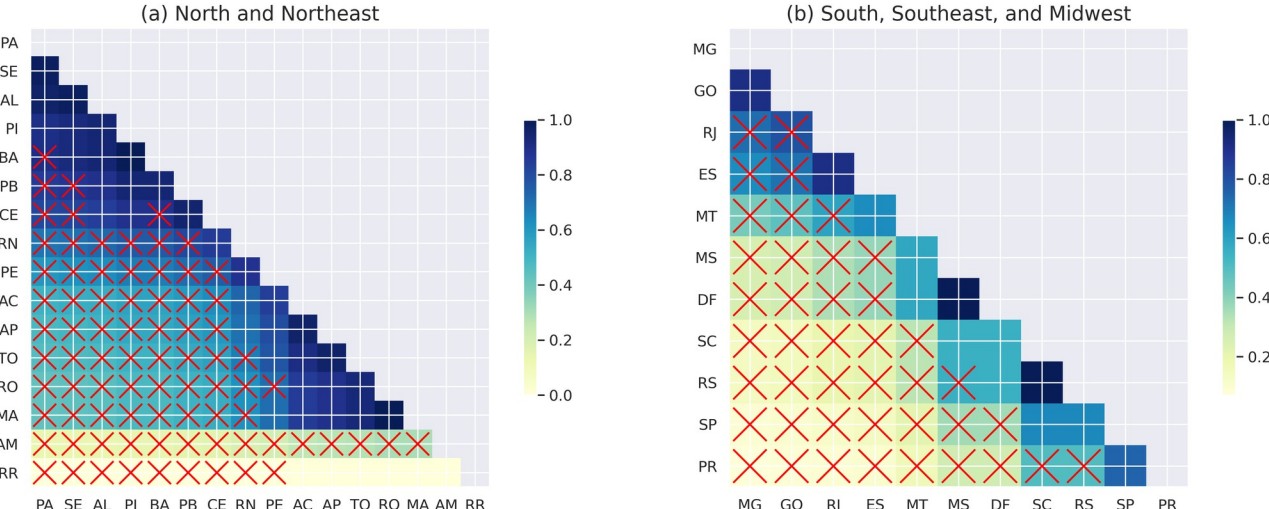

**Fig 6. Illustration of state effectiveness benchmark for the *full* model for the year 2019.** The states are present in the axis in the descendent order for the FNR. Each cell represents the ratio of regions placed on axis x by those on axis y. (a) North and Northeast and (b) South, Southeast and Midwest.

from the North and place the first position. Among the northeastern states, MA and PE have the worst results. MA is the northeastern state with the lowest fraction of positive schools, thereby with more chances to achieve higher FNR being the most effective if a naive model misclassifies positive schools equally (same amount across states). However, MA has the lowest FNR in the Northeast with statically significant differences when compared with all others in the same region. This shows the ability of the proposed approach to untangle unobserved effects present in the data and suggests the relative ineffectiveness of MA policies to overcome poor performance at the school level.

On the other hand, the northeastern state SE, even with a higher fraction of positively labeled schools, achieved the second largest FNR. This suggests that schools that score below the median in SE are relatively more similar to those that score above the median nationally, placing SE as the most effective northeastern state with no statical differences compared with AL and PI. The higher effectiveness of SE partially aligns with their observed score gains at the ENEM, which had only 0.37 schools in the top quartile in 2009 and 0.69 in 2019 (*Fig 4*). *Fig 6(B)* illustrates the remaining regions, with the southeastern and midwestern states generally more effective than the southern states. The MG gets the first position, and the least effective state is PR.

When comparing the results of the *full* model with those of the model that excludes non-actionable variables (*race* and *gender*), we see a decrease in the PA ranking from first to eighth position (Fig 7(A)). In addition, there is a significant difference in the MA ranking, which increases by four positions, although it remains one of the least effective states in the northeast. Fig 7(B) shows that the most significant changes among the southeastern, midwestern, and southwestern states from the *full* model are the MT, which moves up five positions, and the SP, which drops six positions. These differences in results seem more related to the *race* variable than to gender, as race is significantly related to performance in this dataset [44] and race composition varies considerably across Brazil.

To demonstrate the potential of this paper as a complementary strategy to ranking educational systems, Table 6 describes the rank of the top 10 states with the gross score beside the reframed ranking when accounting for effectiveness. Both metrics refer to the average of the 2009–2019 period, and the effectiveness is related to the bias regarding FNR to the *full* model. The metrics were not paired tested.

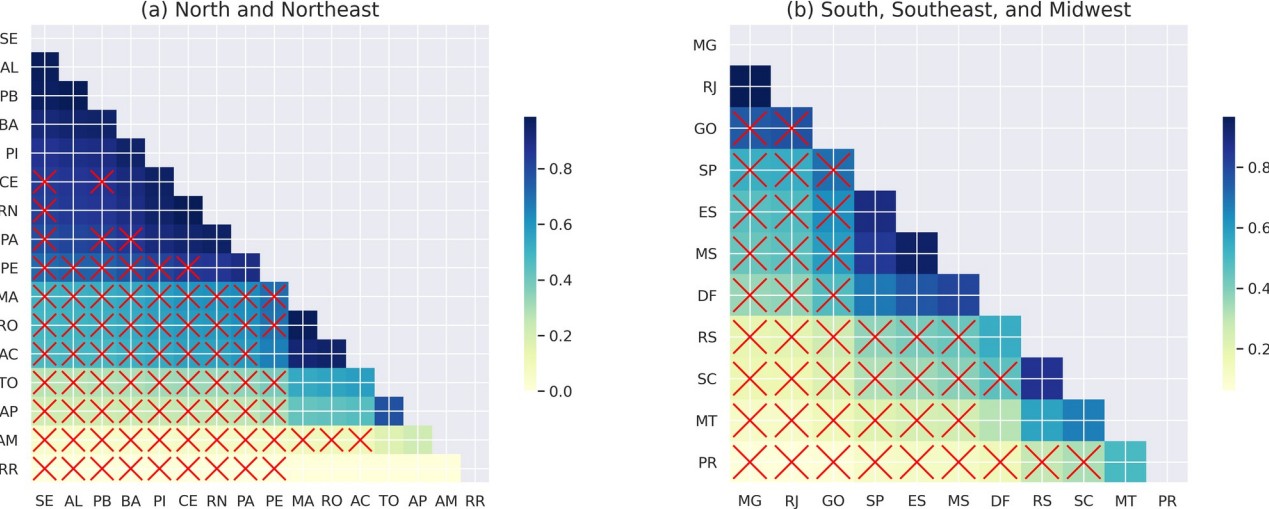

**Fig 7. Illustration of state effectiveness benchmark for the model *school+student+teacher* for the year 2019.** The states are present in the axis in the descendent order for the FNR. Each cell represents the ratio of regions placed on axis x by those on axis y. Clearer colors represent higher absolute difference while the crossed line represents whether they are statistically significant. (a) North and Northeast and (b) South, Southeast and Midwest.

Overall, the effectiveness metrics contrast with the raw scores. Although it is not possible to verify the real value of effectiveness, the new ranking is interesting since it features ES and GO, which had the most significant increases in ranking and, at the same time, achieved the highest gain nationally in LSA gross score from 2009 to 2019, as illustrated in *Fig 4*. Also, it is worth noting that the high effectiveness of these states regarding policies and practices also conforms to earlier results [4] which used data from Brazilian elementary schools from 2007 to 2017 using residuals of linear models in longitudinal data.

**5.2.3 Specific policies.** Following the strategy of contrasting the error performance from the *full* model to the model without a specific set of variables to understand the role of states regarding particular policies, the *full* model was contrasted with those without *students*, *faculty*, *non-actionable*, and *schools*.

All states have statistical significance results from the *full* model to the model without student variables (*school + teacher + non-actionable*). However, only CE in 2015, 2016, 2017, and 2018 did not increase its FNR. In other words, the *student* variables increase considerably the model underestimation for all states except for CE. It could suggest that CE has good

**Table 6. Top 10 average ENEM scores from 2009 to 2019 and effectiveness ranking based on the FNR.** The position change is assigned, being "+" an ascendant behavior and "–"a descendent.

|  | Gross Score | Effectiveness (FNR) |
|---|---|---|
| 1 | DF | MG (+2) |
| 2 | RS | RJ (+3) |
| 3 | MG | GO (+7) |
| 4 | SC | ES (+5) |
| 5 | RJ | MS (+3) |
| 6 | SP | DF (-5) |
| 7 | PR | RS (-5) |
| 8 | MS | SP (-2) |
| 9 | ES | SC (-5) |
| 10 | GO | PR (-3) |

educational policies to alleviate the effect of students' backgrounds compared to others. However, the potential high correlation between *student variables and the remaining others in the model*, *such as non-actionable* and *school* characteristics, requires caution in this interpretation and more investigation, since these variables could serve as a proxy for the bias from *student* features in CE more than in the rest of the country.

Taking off the *non-actionable* variables, RS [Rio Grande do Sul] is highlighted as the state that that saw a significant reduction in FPR in nine years. In general, there are no significant differences in the other states, except in SP for two years and SC for just one year. Therefore, when ignoring *non-actionable* variables (*race + gender*), RS stands out as the state that saw the largest decrease in school overestimation. However, this observation does not provide any insights into policies as it is heavily expected. RS has the highest concentration of white people, 82% in 2019, while the national average was 42%. White people are strongly linked to high academic achievement in Brazil when analyzing the entire country [45]. The strong positive influence of the white race and the high concentration of white students in RS lead this state to have these significant differences in misclassifications of educational effectiveness when this variable is present in the model and when it is not.

Nevertheless, BA, which has the lowest concentration of white people (15%), as opposed to RS with a high value of FNR, also should present a significant decrease in FNR when the model becomes blind to *race*. However, interestingly, as the other states, BA has no statistically significant changes in FNR. This behavior suggests a certain robustness in BA´s policies regarding non-actionable variables (*race + gende*r). More specifically, regarding *race*, as gender has low relevance in discriminating school achievement in Brazilian secondary education [46], and BA has a highly similar rate for the gender variable (62%), close to the national average (60%). It is worth noting that this BA equity regarding race does not mean that students of BA score higher than the same race groups in other states, but only that race is relatively less important in BA when compared with other states.

Regarding the set of *school* variables, only RR in 2009 presented a significant decrease in its FNR when this set of variables were removed, suggesting that RR schools could overcome poor school characteristics relatively better than the rest in 2009. On the other hand, PR is the only state that had an increased FPR in the years 2009, 2016, and 2017. It suggests that low-achievement schools in PR have relatively more similar infrastructure to high-achievement schools from a national perspective. Therefore, PR cannot fully convert these good conditions into performance when compared with other states. The same happens to *faculty* variables, and PR is the only state with increased FPR. Regarding the FNR, the states of PI, BA, and RR had significant increases in 2011, 2013, and 2014 respectively. In other words, high-achieved schools in these states have increased their tendency to be classified as a low achievement when considering faculty information, which does not happen in other states. Therefore, suggesting relatively good policies regarding teachers.

The statistical tests to compare the *full* model with the models without an underlying set of variables were computed independently for the FNR and FPR for models using the same algorithm. Different algorithms can use totally differently the same set of variables in the same training data to reach similar results [47], and the comparison of ΔFPR and ΔFNR across different algorithms cannot be appropriate. In this case. The results from the RF models were chosen since RF attained a higher performance (F1) for the *full* model, as described in Table 2.

**5.2.4 Over time.** In this section, we analyze the state's effectiveness over time. Although evaluating the evolution of effectiveness itself is impossible since the model was performed independently without considering temporal relationships, the temporal perspective is important to understand how different the states have been highlighted in the whole period.

**Table 7. The five states with the highest FNR that have at least 20% of positive class and a maximum of 80% yearly.**

|  | 2009 | 2010 | 2011 | 2012 | 2013 | 2014 | 2015 | 2016 | 2017 | 2018 | 2019 |
|---|---|---|---|---|---|---|---|---|---|---|---|
| 1˚ | CE | PA | BA | CE | CE | CE | CE | CE | CE | SE | SE |
| 2˚ | PA | PE | PE | BA | PB | SE | PB | SE | SE | PB | PB |
| 3˚ | PB | MG | MG | PE | SE | PB | SE | PB | PB | CE | PI |
| 4˚ | BA | RJ | ES | MG | PA | BA | PA | PA | PA | BA | CE |
| 5˚ | SE | GO | GO | ES | BA | PE | BA | BA | BA | RN | PA |

From the demographic model (*student + non-actionable*), it is possible to evaluate the role of states in all other variables exogenous to student information, including teacher and school characteristics. *Table 7* describes the top underrated states in relation to the rest of the country in the whole period. For a more reliable comparison, it was only included in this analysis the states with a minimum of 20% of the positive class and a maximum of 80%.

The behavior of CE is interesting since it is indicated as the first state to perform better than expected in seven years and has been in the top five for nine years. This result is in line with educational literature, which has already highlighted the high effectiveness of this state in promoting basic education and overcoming adverse socioeconomic conditions based on its practices [48, 49]. To make it straightforward, Table 8 describes a score order to states analyzed when weighing the occurrence of each state in an underlying position. The first position received a weight of five, while the following positions decreased until the fifth received a weight of one.

## 5.3 Traditional model

*Fig 8* shows the state's *random effect* ranking with their respective confidence interval for 2019. The *random effects* are the variance of the state dummy variable computed 'for the same HLM using all features without the *non-actionable* features set. Although a complete comparison of models is not possible since they are based on different outputs, the models have several similarities when compared with the estimated results illustrated in *Fig 6*. For instance, both highlight the high effectiveness of northeast states, such as SE, PB, CE, PI, and BA, and those of the southeast, such as MG and RJ. Also, MA is the lowest effective northeastern state for both models.

**Table 8. Weighted score of the five states with the highest FNR metric with at least 20% and a maximum of 80% of schools with an average ENEM score above the median.** The score is computed using the number of occurrences in an underlying position from 2009 to 2019 multiplied by the weight relative to each position. Five is the weight for the first and one for the fifth.

| State | Occurrences*weight |
|---|---|
| CE | 848 |
| SE | 325 |
| BA | 154 |
| PA | 153 |
| PB | 120 |
| PE | 55 |
| MG | 14 |
| PI | 6 |
| ES | 3 |
| RJ | 2 |
| GO | 2 |
| RN | 1 |

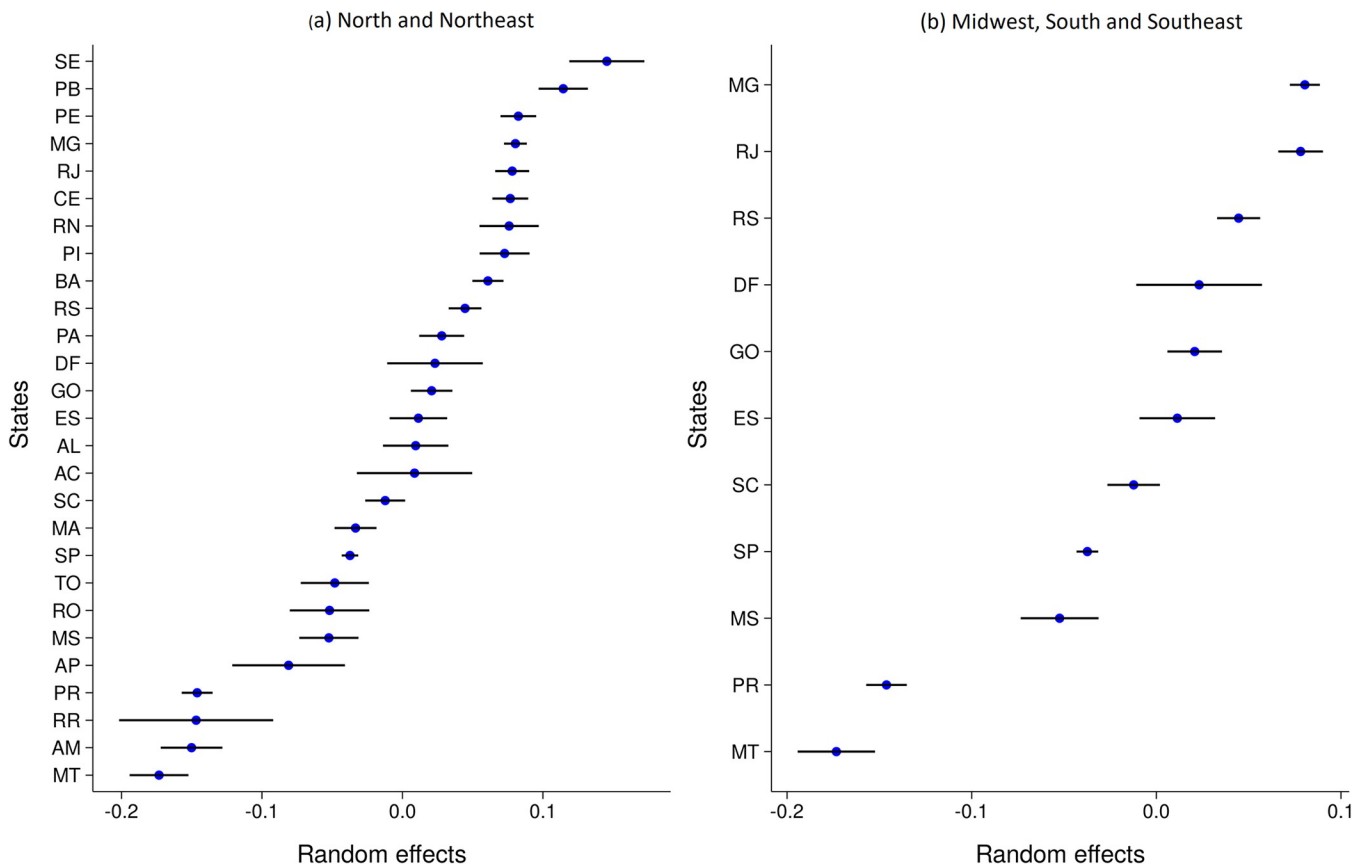

**Fig 8. Random effects of Brazilian states grouped by regions with similar scores for the year 2019 with their respective confidence interval computed in the same HLM model.** (a) North and Northeast and (b) Midweast, South and Southeast.

Furthermore, states identified as low effective in the previous estimates, such as RR, PR, AM, TO, and MT, are characterized by a negative *random effect*. Overall, the models do not have significant divergence. The AL and GO was shown to have higher effectiveness in the proposed approach but had a lower performance in the HLM models, although still positive. The opposite holds for PE, which was highlighted in the HLM model and assigned with a lower relevance in the new approach.

## 6. Discussion

A critical exercise in LSA data analysis is identifying effective educational systems. This identification allows further in-depth investigation into these systems to shed light on possible practices and policies that lead to educational improvement. To explore the growing range of information derived from modern LSA, new techniques have been proposed [9] to add to the traditional statistical tools [4, 15, 50]. This paper aims to contribute to this literature by presenting a new approach based on the ML-supervised learning paradigm. The use of ML models is useful to avoid model misspecification since they are able to capture more variance in the data by adapting the predictive function in a data-driven approach. Therefore, considering a model with low variance, the bias of model can be more accurately assigned to the differences among administrative units. The main difference between the present study and previous ones is our ability to explore extensive datasets accounting for

many variables without making assumptions about the data-generating process. Also, the new approach allows a fine-grain and paired analysis which opens the path to more realistic comparisons. Finally, it is model-independent, allowing the use of an increasing range of machine learning (ML) classifiers.

The estimates we make for the Brazilian secondary education context aim to identify how different education is delivered across the country. This analysis adds a new contribution to this paper since, to our knowledge, these data have not been explored to compare effectiveness. Additionally, traditional hierarchical linear models (HLM), the most common approach used for this purpose, was carried out for comparative purposes. The results align with existing educational literature and could explain some differences in LSA scores. Overall, the approaches show a high similarity among the more and less effective educational systems when compared for the same year using the same set of variables.

Additional findings of the estimates, such as the effectiveness of some states in overcoming some problems such as racial inequality (BA), poor school infrastructure (RR), and teacher limitation (PI, BA, and RR), can inform policymakers or motivate further investigations in these states to invest resource more efficiently. Also, some states deserve more attention, such as PR, which has taken relatively little advantage of its good school and teachers' characteristics. Finally, MA, with lower overall performance, seems not to be able to overcome its limitations, and it is one the most ineffective state in almost the whole period studied.

Nevertheless, despite the similar results to the HLM analysis, the interpretation of results always requires caution and further in-depth investigation. The analysis was performed without taking into consideration critical factors that could impact our results. For instance, we use the average performance as a start-point for the effectiveness benchmark. However, the average score can be positively influenced by the low effectiveness of administrative unit policies. This contradiction can lead to bias in the results. The number of students that attend ENEM has been decreasing over the years, and this rate can be highly correlated with drop-off in schools. A state with low effectiveness in maintaining its most vulnerable students in schools may see its average score increase, not due to the performance of the remaining students, but rather due to a lack of policies in place to support the most vulnerable students. Therefore, the analysis conducted in this paper assumes that students who attend and do not attend ENEM have similar biases regarding all variables included in the model. Also, as a cross-sectional analysis, the effectiveness is not the same as value-added, extensively explored in the educational literature using longitudinal data. In this current study, the effectiveness is only the difference in the bias of models when trained with a different set of features. This analysis is very similar to the residual analysis or HLM, but with fewer assumptions about the data generation process in their relationships.

Future works should extend these present evaluations by considering different options. For instance, a resampling design can leverage the confidence of results and increase the range of fair comparison since class imbalance is a significant limitation for this study. Also, a longitudinal analysis that benefits from the temporal structure of the dataset can be used to investigate how effectiveness changes over time. Finally, the application of this approach to other LSA considering other countries or regions can be useful to understand the generalizability of the proposed method.

It is hoped that this approach will prove useful not only for researchers in educational assessment but also in other fields that seek to understand within-group differences, such as in the healthcare and social science fields. For example, this approach can be used to verify the effectiveness of public policies in other contexts, such as the operational effectiveness of hospitals that share similar treatments but with different infrastructure and human resources.

## Supporting information

**S1 Appendix.**
(DOCX)

## Author Contributions

**Conceptualization:** Rogério Luiz Cardoso Silva Filho, Kellyton Brito, Paulo Jorge Leitão Adeodato, Martin Carnoy.

**Data curation:** Rogério Luiz Cardoso Silva Filho.

**Formal analysis:** Rogério Luiz Cardoso Silva Filho.

**Investigation:** Rogério Luiz Cardoso Silva Filho, Martin Carnoy.

**Methodology:** Rogério Luiz Cardoso Silva Filho, Kellyton Brito, Paulo Jorge Leitão Adeodato, Martin Carnoy.

**Software:** Rogério Luiz Cardoso Silva Filho.

**Supervision:** Kellyton Brito, Paulo Jorge Leitão Adeodato, Martin Carnoy.

**Validation:** Rogério Luiz Cardoso Silva Filho, Anvit Garg, Kellyton Brito, Paulo Jorge Leitão Adeodato.

**Visualization:** Rogério Luiz Cardoso Silva Filho.

**Writing – original draft:** Rogério Luiz Cardoso Silva Filho, Kellyton Brito, Paulo Jorge Leitão Adeodato, Martin Carnoy.

**Writing – review & editing:** Rogério Luiz Cardoso Silva Filho, Anvit Garg, Kellyton Brito, Paulo Jorge Leitão Adeodato, Martin Carnoy.

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
