## [Decision Letter · Decision Letter 0]

5 Jun 2023

PONE-D-23-04852A MACHINE LEARNING APPROACH TO COMPARING EDUCATIONAL EFFECTIVENESS IN A LARGE DEVELOPING COUNTRY, BRAZILPLOS ONE

Dear Dr. Cardoso Silva Filho,

Thank you for submitting your manuscript to PLOS ONE. After careful consideration, we feel that it has merit but does not fully meet PLOS ONE’s publication criteria as it currently stands. Therefore, we invite you to submit a revised version of the manuscript that addresses the points raised during the review process.

We look forward to receiving your revised manuscript.

Kind regards,

José S. Andrade Jr.

Academic Editor

PLOS ONE

Journal Requirements:

Reviewers' comments:

Reviewer's Responses to Questions

**Comments to the Author**

1. Is the manuscript technically sound, and do the data support the conclusions?

Reviewer #1: Yes

2. Has the statistical analysis been performed appropriately and rigorously? 

Reviewer #1: Yes

3. Have the authors made all data underlying the findings in their manuscript fully available?

Reviewer #1: Yes

4. Is the manuscript presented in an intelligible fashion and written in standard English?

Reviewer #1: Yes

5. Review Comments to the Author

Reviewer #1: This paper provides an approach based on supervised machine learning models that the authors claim is flexible, model independent, and that allows more realistic comparisons. The authors apply their approach to data from the Brazilian Educational Exam, where they intend to verify the suitability of the method to explore differences in effectiveness between Brazilian educational administrative units at different regional levels over a period of 10 years. According to the authors, this approach allows exploring extensive datasets without making assumptions about the data generation process, in the case of the Brazilian Educational Exam they also claim to produce a number of new discoveries not observed in traditional approaches.

Overall, the paper is well written. It brings contributions to the use of machine learning techniques as a research tool for the improvement of public policies. Thus, it seems appropriate for PLOS ONE. However, there are some points that must be addressed:

1 - I suggest that the authors standardize the tables in the manuscript, as well as increase the resolution of the figures to improve the readability of the work. In addition, the figure 5 is with the labels of the regions in Portuguese, unlike what happens in the other figures.

2 - It is unclear which fractions of the dataset the authors used for training/testing, as well as whether a sampling strategy was used to ensure a good fit of the models. This is important to facilitate the reproducibility of the results presented in the work. Therefore, I suggest that the authors clarify this in the text.

3 - In section "4.5 Modeling and evaluation" the authors consider the use of three classifiers: logistic regression, random forest and adaboosting. Did the authors adjust the hyperparameters for these models? It would be important to make clear which hyperparameters were varied or which values were set if they were kept fixed.

6. PLOS authors have the option to publish the peer review history of their article (what does this mean?). If published, this will include your full peer review and any attached files.

Reviewer #1: No

---

## [Author Response · Author response to Decision Letter 0]

7 Jun 2023

Response to Reviewers

R1Q1 - I suggest that the authors standardize the tables in the manuscript, as well as increase the resolution of the figures to improve the readability of the work. In addition, the figure 5 is with the labels of the regions in Portuguese, unlike what happens in the other figures.

R. Thanks for the comment. The tables were standardized under a simple layout, and the figures are present in 400 DPI resolution. Also, the Figure 5 labels were updated.

R1Q2 - It is unclear which fractions of the dataset the authors used for training/testing, as well as whether a sampling strategy was used to ensure a good fit of the models. This is important to facilitate the reproducibility of the results presented in the work. Therefore, I suggest that the authors clarify this in the text.

R. Thanks for this comment. The information about the fraction used to train and test the algorithm is explained in the second paragraph of subsection 4.5. To make it clearer, we have updated this paragraph. 

R1Q3 - In section "4.5 Modeling and evaluation," the authors consider the use of three classifiers: logistic regression, random forest and adaboosting. Did the authors adjust the hyperparameters for these models? It would be important to make clear which hyperparameters were varied or which values were set if they were kept fixed.

R. Thanks for this comment. This information is already presented in the last sentence of the last paragraph in section 4.5. We have used the default parametrization. However, we have updated this paragraph to justify that.

---

## [Decision Letter · Decision Letter 1]

17 Jul 2023

A MACHINE LEARNING APPROACH TO COMPARING EDUCATIONAL EFFECTIVENESS

PONE-D-23-04852R1

Dear Dr. Cardoso Silva Filho,

We’re pleased to inform you that your manuscript has been judged scientifically suitable for publication and will be formally accepted for publication once it meets all outstanding technical requirements.

Kind regards,

José S. Andrade Jr.

Academic Editor

PLOS ONE

Additional Editor Comments (optional):

Reviewers' comments:

Reviewer's Responses to Questions

**Comments to the Author**

1. If the authors have adequately addressed your comments raised in a previous round of review and you feel that this manuscript is now acceptable for publication, you may indicate that here to bypass the “Comments to the Author” section, enter your conflict of interest statement in the “Confidential to Editor” section, and submit your "Accept" recommendation.

Reviewer #1: All comments have been addressed

2. Is the manuscript technically sound, and do the data support the conclusions?

Reviewer #1: Yes

3. Has the statistical analysis been performed appropriately and rigorously? 

Reviewer #1: Yes

4. Have the authors made all data underlying the findings in their manuscript fully available?

Reviewer #1: Yes

5. Is the manuscript presented in an intelligible fashion and written in standard English?

Reviewer #1: Yes

6. Review Comments to the Author

Reviewer #1: All comments and suggestions have been duly addressed and the article is ready for publication in PLOS ONE.

7. PLOS authors have the option to publish the peer review history of their article (what does this mean?). If published, this will include your full peer review and any attached files.

Reviewer #1: No

---

## [Editor Report · Acceptance letter]

19 Jul 2023

PONE-D-23-04852R1 

Beyond scores: A machine learning approach to comparing educational system effectiveness 

Dear Dr. Cardoso Silva Filho:

I'm pleased to inform you that your manuscript has been deemed suitable for publication in PLOS ONE. Congratulations! Your manuscript is now with our production department. 

Kind regards, 

on behalf of

Prof José S. Andrade Jr. 

Academic Editor

PLOS ONE